

# Identification and validation of a ferroptosis-related lncRNA signature to robustly predict the prognosis, immune microenvironment, and immunotherapy efficiency in patients with clear cell renal cell carcinoma

Lincheng Ju, Yaxing Shi and Gang Liu

Department of Urology, Shengjing Hospital of China Medical University, Shenyang, China

## ABSTRACT

**Background**. Ferroptosis is a new type of iron- and reactive oxygen species-dependent cell death, studies on ferroptosis-related long noncoding RNAs (FerLncRNAs) in clear cell renal cell carcinoma (ccRCC) are limited. The purpose of this study was to investigate the potential prognostic value of FerLncRNAs and their relationship with the immune microenvironment and immunotherapy response of ccRCC.

**Methods**. RNA sequencing data of 526 patients with ccRCC were downloaded from The Cancer Genome Atlas (TCGA) database. The patients with ccRCC in TCGA were randomly divided (1:1) into a training and testing cohort. ICGC and GEO databases were used for validation. Screening for FerLncRNAs was performed using Pearson's correlation analysis with the reported ferroptosis-related genes. A FerLncRNA signature was constructed using univariate, LASSO, and multivariate Cox regression analyses in the training cohort. Internal and external datasets were performed to verify the FRlncRNA signature. Four major FRlncRNAs were verified through *in vitro* experiment.

**Results**. We identified seven FerLncRNAs (LINC00894, DUXAP8, LINC01426, PVT1, PELATON, LINC02609, and MYG1-AS1), and established a risk signature and nomogram for predicting the prognosis of ccRCC. Four major FRlncRNAs were verified with the prognosis of ccRCC in the GEPIA and K-M Plotter databases, and their expressions were validated by realtime PCR. The risk signature can also effectively reflect the immune environment, immunotherapy response and drug sensitivity of ccRCC. These FRlncRNAs have great significance to the implementation of individualized treatment and disease monitoring of ccRCC patients.

## INTRODUCTION

Renal cell carcinoma (RCC) is the ninth most frequent cancer in males and fourteenth most frequent cancer in females, the rise in RCC morbidity and mortality around the

Corresponding author
Gang Liu, gangliu_sj@126.com

world has raised serious concerns (*Nabi et al., 2018*). The median overall survival (OS) rate of patients with RCC remains unsatisfactory despite the recent rapid discovery and extensive clinical usage of antiangiogenic drugs and immune checkpoint inhibitors (ICIs) for the treatment of cancer, including RCC (*Chen et al., 2020*). Therefore, it is crucial to investigate the the pathophysiology and development mechanisms of RCC and explore novel therapeutic targets. Clear cell RCC (ccRCC) is the most common type of RCC, accounting for approximately 75–80% of all cases of RCC (*Nabi et al., 2018*). Recent research has demonstrated that ccRCC cells are vulnerable to ferroptosis, a new form of iron-dependent programmed death that differs from previous types of cell death in terms of morphologically, genetically, and chemically properties (*Chen et al., 2020*; *Ganini et al., 2022*). Lipid peroxidation, iron buildup, and glutathione deprivation are the hallmarks of ferroptosis, which is intimately associated with the formation and progression of cancer (*Stockwell et al., 2017*). A unique approach to treat cancer, particularly in cases of drug resistance following traditional therapy, involves activating the process of ferroptosis in cancer cells (*Chen et al., 2020*).

Long noncoding RNAs (lncRNAs) are noncoding RNAs greater than 200 nucleotides in length. Genetic research has turned its attention to lncRNAs because studies have revealed that these molecules are crucial for the control of the cell cycle, differentiation, and epigenetic regulation (*Zhai et al., 2019*). Although lncRNAs do not encode proteins, they have several specific functions, such as transcriptional regulation, mRNA processing, and posttranscriptional regulation of mRNA (*Zhai et al., 2019*). Studies have also shown that lncRNAs play a role in regulating ferroptosis (*Mao et al., 2018*; *Xie & Guo, 2021*); however, the impact of ferroptosis on lncRNAs-dependent ccRCC progression, immune microenvironment and immunotherapy response has been scarcely studied (*Han et al., 2022*; *Xing et al., 2021*; *Zhou et al., 2022*).

In this study, we aimed to investigate the ferroptosis-related genes (FRGs) and FerLncRNAs in ccRCC. We established a ferroptosis-related lncRNA prognostic signature to predict the individual prognosis of ccRCC. To validate the prognostic signature, we investigated its efficiency and accuracy in the training, testing, total cohorts in TCGA database, as well as ICGC database. The results obtained with GEPIA, and K-M Plotter data supported the predictive ability of the major lncRNAs in risk signature. In addition, immune cell infiltration and check point expressions associated with this signature were explored. Our study demonstrated and proved that the prognostic signature can be applied in the clinical prognosis of ccRCC patients. Importantly, our study provided a new approach for predicting the response to treatment, including immunotherapy, chemotherapy and targeted therapy, in ccRCC patients.

## MATERIALS AND METHODS

### Data source
The RNA-seq (FPKM) data for ccRCC samples ($n = 539$) and adjacent nontumorous kidney samples ($n = 72$) were retrieved from The Cancer Genome Atlas (TCGA) (https://portal.gdc.cancer.gov). Corresponding clinicopathological characteristics of

**Table 1  Clinical characteristics and related cohort grouping of 526 patients with ccRCC.**

| Variates | Type | Overall cohorts (n = 526) | Training cohorts (n = 264) | Testing cohorts (n = 262) | P value |
|---|---|---|---|---|---|
| Age | <=60 years | 264(50.19%) | 135(51.14%) | 129(49.24%) | 0.7275 |
| | >60 years | 262(49.81%) | 129(48.86%) | 133(50.76%) | |
| Gender | Female | 183(34.79%) | 97(36.74%) | 86(32.82%) | 0.3944 |
| | Male | 343(65.21%) | 167(63.26%) | 176(67.18%) | |
| Grade | G1–G2 | 239(45.44%) | 124(46.97%) | 115(43.89%) | 0.631 |
| | G3–G4 | 279(53.04%) | 137(51.89%) | 142(54.2%) | |
| | Unknown | 8(1.52%) | 3(1.14%) | 5(1.91%) | |
| Stage | I–II | 318(60.46%) | 164(62.12%) | 154(58.78%) | 0.4872 |
| | III–IV | 208(39.54%) | 100(37.88%) | 108(41.22%) | |
| M | M0 | 418(79.47%) | 207(78.41%) | 211(80.53%) | 0.0709 |
| | M1 | 78(14.83%) | 36(13.64%) | 42(16.03%) | |
| | Unknown | 30(5.7%) | 21(7.95%) | 9(3.44%) | |
| N | N0 | 238(45.25%) | 117(44.32%) | 121(46.18%) | 0.9085 |
| | N1 | 16(3.04%) | 8(3.03%) | 8(3.05%) | |
| | Unknown | 272(51.71%) | 139(52.65%) | 133(50.76%) | |
| T | I–II | 336(63.88%) | 177(67.05%) | 159(60.69%) | 0.1535 |
| | III–IV | 190(36.12%) | 87(32.95%) | 103(39.31%) | |

patients with ccRCC ($n = 526$), including OS, were also obtained from the TCGA database. Based on the patient ID number, we matched their transcriptome data with clinical information and excluded the data for patients who did not match. Thus, we obtained complete gene expression profiles of 526 patients with ccRCC. Using the caret R package, all patients with ccRCC were randomly divided into two cohorts (1:1): training and test. The specific clinical parameters for both cohorts, as well as for the entire TCGA cohort, are shown in Table 1. There were no significant differences in the clinical characteristics of patients between the training and test cohorts.

From the Gene Expression Omnibus (GEO) database (http://www.ncbi.nlm.nih.gov/geo/), three ccRCC datasets (GSE15641 (*Jones et al., 2005*), GSE46699 (*Eckel-Passow et al., 2014*), and GSE40435 (*Wozniak et al., 2013*)) were selected for bioinformatics analysis. The expression data of 91 ccRCC patients from the ICGC database (https://dcc.icgc.org/analysis) were obtained for the external validation of the risk signature.

## Identification of FerLncRNAs

lncRNAs and protein-coding genes were classified using the Ensembl human genome browser GRCh38.p13 (http://asia.ensembl.org/index.html) (*Cunningham et al., 2019*). FRGs were identified in the FerrDb (http://www.zhounan.org/ferrdb/current/) (*Zhou & Bao, 2020*). Pearson's correlation coefficients were then calculated to determine the correlation between the expression of FRGs and the corresponding lncRNAs. Ferroptosis-associated lncRNAs were identified based on $p < 0.001$ and the absolute value of the Pearson's correlation coefficient greater than 0.35.

## Construction and validation of a prognostic FerLncRNA signature

Screening for differentially expressed FRGs and lncRNAs between tumor and nontumor tissues was performed using the edgeR R package. The screening criteria were a false discovery rate of $p < 0.001$ and $|\log_2 FC| \geq 2$, LncRAN with expression value of 0 was deleted. Next, the differentially expressed lncRNAs were analyzed by univariate Cox regression analysis with OS to identify the prognostic lncRNAs in the entire cohort. The univariate variables with $P$-values of <0.05 were included in the least absolute shrinkage and selection operator (LASSO) analysis, which was used to further select useful predictive features to avoid overfitting of the model. Subsequently, a multivariate Cox regression analysis of candidate ferroptosis-associated lncRNAs was performed to assess their contribution as prognostic factors for OS in the training group. Finally, seven best ferroptosis-associated lncRNAs were identified for the prognostic model. The risk score of each patient was calculated based on this prognostic signature according to the normalized expression levels of FerLncRNAs and corresponding regression coefficients as follows:

$$\text{Risk score} = \sum_{k=1}^{n} \text{Coef}\kappa * X\kappa.$$

The patients in the training cohort were divided into low- and high-risk groups based on the median risk score, and OS was compared between the groups using the Kaplan–Meier method and log-rank test. The area under the time-dependent ROC curve (AUC) was then determined using the survivalROC R package to assess the predictive accuracy of lncRNA features associated with ferroptosis. To validate this prognostic model, the risk score for each patient in the test cohorts, overall cohorts, and ICGC cohorts was calculated using the same way to confirm the stability of the established model.

## Kyoto Encyclopedia of Genes and Genomes (KEGG) pathway analysis and Gene Ontology (GO) enrichment analysis of differentially expressed FRGs

The clusterProfiler R package was used for the enrichment analysis of differentially expressed genes based on KEGG and GO pathway annotations.

## Construction of an lncRNA–mRNA coexpression network

To demonstrate the correlation between FerLncRNAs and their corresponding mRNAs, we mapped an mRNA–lncRNA coexpression network using the Cytoscape software (version 3.8.2, Mac OSX, http://www.cytoscape.org/). A Sankey diagram was then plotted to further demonstrate the degree of correlation between FerLncRNAs and their corresponding mRNAs.

## Principal component analyses

We utilized the principal component analysis (PCA) to reduce the dimension and visualize the renal cancer patients with different risk values.

## Immunity analysis and gene expression

Based on the results of immunotyping of pancancer in the literature (*Thorsson et al., 2018*), we compared the relationship between risk score and different immunotyping (C1: wound

healing, C2: IFN-gamma dominant, C3: inflammatory, C4: lymphocyte depleted, C5: immunologically quiet, and C6: TGF-beta dominant-characterized). The infiltration of immune cells was evaluated using the CIBERSORT tool (*Newman et al., 2015*). Immune and stromal cell scores were calculated using the Estimation of Stromal and Immune cells in Malignant Tumor tissues using Expression data (ESTIMATE) algorithm. The Tumor Immune Estimation Resource (TIMER), CIBERSORT, CIBERSORT-ABS, QUANTISEQ, MCPCOUNTER, XCELL, and EPIC algorithms were compared to assess cellular components or cell immune responses between the high- and low-risk groups based on the FerLncRNA signature. In addition, the algorithm for single sample gene set enrichment analysis (ssGSEA) was utilized to determine the scores of tumor microenvironment (TME) cells in each sample (*Rooney et al., 2015*) with the GSVA package. This score is used to quantify the types of tumor infiltrating immune cells between the two groups. Potential immune checkpoints were retrieved from previous literature.

## External verification

The gene expression profiling interactive analysis (GEPIA, http://gepia.cancer-pku.cn/) (*Tang et al., 2017*) contained RNA-seq and clinical data compiled by TCGA and GTEx after standardized analysis. The K-M Plotter database (http://kmplot.com/analysis/; *Lanczky & Gyorffy, 2021*) included data on the correlation between the gene expression and prognostic data of 530 patients with ccRCC. The expression levels of the major FRlncRNAs and the prognostic correlation were verified in the above two online databases.

## Cell culture and treatment

Human renal proximal tubule epithelial cells (HK-2 cells) and renal clear cell carcinoma cell lines (786-0, and Caki-1) were obtained from the American Type Culture Collection (ATCC, Manassas, VA, United States). Cells were incubatedat 37 °C in a humid 5% $CO_2$ environment and routinely cultured in RPMI 1640 or DEM supplemented with 10% fetal bovine serum (Invitrogen, Carlsbad, CA, United States).

## RNA extraction, reverse transcription, and quantitative real-time PCR (qRT-PCR)

A spectrophotometer is used to determine the quantity and quality of total RNA after it has been extracted from the aforementioned cells using the total RNA extraction micro-Kit (RNT411-03; Guangdong, China). Then, cDNA was synthesized using SuperScript II Reverse Transcriptase, oligo 18dT, and random primers (hexamers) (Invitrogen). On a Roche LightCycler 480 sequence detection system, qRT-PCR was carried out under the following conditions: 30 s of predenaturation at 95 °C, 40 cycles of denaturation at 95 °C lasting 5 s, and 30 s of annealing and extension at 60 °C. The constitutive control used was human glyceraldehyde-3-phosphate dehydrogenase (GAPDH), and the levels of gene expression were determined using the $2^{-\Delta\Delta}$ CT function.The following

primer sequences were used: PVT1, (forward) 5′-CCTGGTGAAGCATCTGATGCACG-3′ and (reverse) 5′-GCCAGGCTTTGTGGCACACGC-3′; LINC00894, (forward) 5′-GCAGGGTCTCTTGAGTTCCT-3′, and (reverse) 5′-TTCCTCAAGCTTCTCCAGGG-3′; DUXAP8, (forward) 5′-AGGATGGAGTCTCGCTGTATTGC-3′, and (reverse) 5′-GGAGGTTTGTTTCTTCTTTTTT-3′; LINC01426, (forward) 5′-CGCACCCAGATACTTTTCGT-3′, and (reverse)5′-GCCGTTGAGGTTGTCGTAAT-3′. PCR reactions of each sample were done in triplicate. All results are presented as the mean ± standard deviation (SD).

## Prediction of immunotherapy response

Tumor Immune Dysfunction and Exclusion (TIDE, http://tide.dfci.harvard.edu/) (*Fu et al., 2020*) algorithms were used to predict immune checkpoint response inhibitors in the high-risk and low-risk group. Immune Cell Abundance Identifier (ImmuCellAI, http://bioinfo.life.hust.edu.cn/ImmuCellAI#!/) (*Miao et al., 2020*) is a computational method published in 2020 to predict the response to immune checkpoint blockade based on the abundance of immune cells, particularly different T cell subsets. The abundance of infiltrating immune cells was calculated by ImmuCellAI and used to develop the response prediction model. The immunotherapy response prediction model was developed using a support vector machine with the radial basis function kernel.

## Assessment of the sensitivity of chemotherapy and molecular drugs

To estimate the risk score in predicting the response to chemotherapy and molecular drugs, the pRRophetic R package was applied to calculate the half-maximal inhibitory concentration (IC50) of samples between the low-risk and high-risk groups. The IC50 between the low- risk and high-risk groups was compared by the Wilcoxon signed-rank test.

## Statistical analysis

The Wilcoxon test was used to compare the proportions of tumor-infiltrating immune cells and the expression levels of immune checkpoint molecules between the high- and low-risk groups. Spearman's correlation analysis was used to analyze the correlations between tumor-infiltrating immune cells. Differences in the proportions of clinical features were analyzed using a chi-squared test. Univariate Cox regression, LASSO, and multivariate Cox regression analyses were performed to determine independent prognostic factors for OS. The predictive accuracy of the OS prognostic model was assessed by a time-dependent AUC analysis. All statistical analyses were performed using the R software (version 4.1.0; *R Core Team, 2021*) and RStudio (version 2021.09.1 Build 372 for macOS; *RStudio Team, 2021*). Statistical significance was defined as a two-tailed *P*-value of <0.05.

## RESULTS

### Identification of prognostic differentially expressed FerLncRNAs in Patients with ccRCC

The overall workflow of this study is shown in Fig. S1. Analysis of RNA-seq data from patients with ccRCC resulted in the identification of 14,056 lncRNAs. FerLncRNAs were

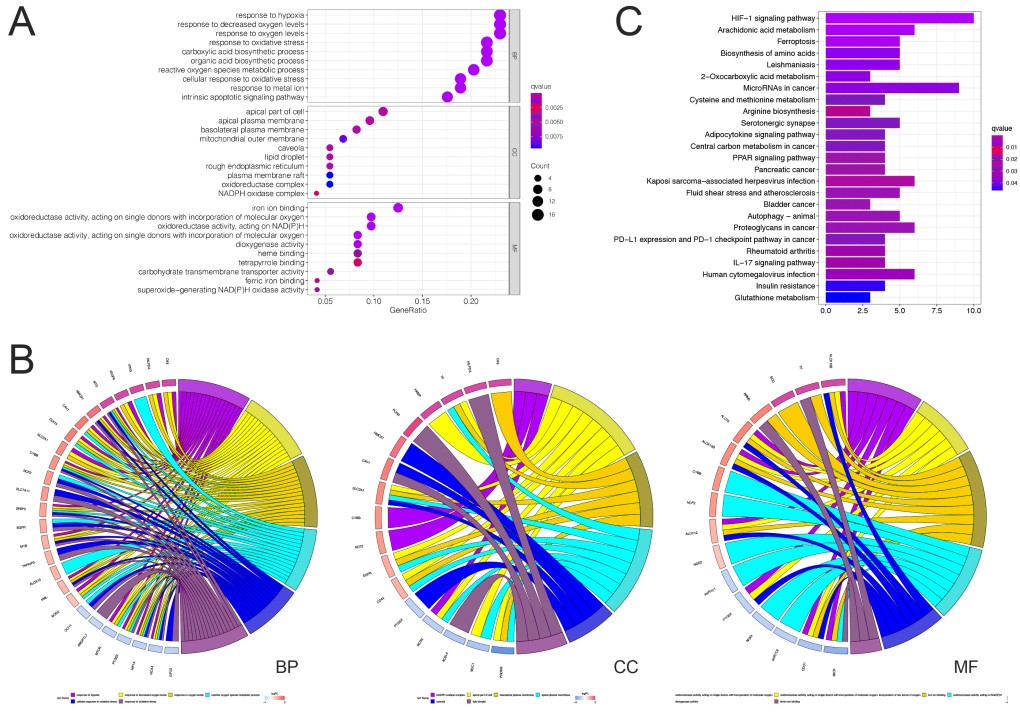

**Figure 1  GO and KEGG pathway analyses of differentially expressed FRGs in ccRCC.** (A) GO analysis of differentially expressed FRGs. (B) Chord plots illustrating the top six enriched items in the GO terms biological process, cellular component, and molecular function (right); genes contributing to the enrichment (left), arranged by their expression levels. (C) KEGG pathway analysis of differentially expressed FRGs.

identified using 259 downloaded FRGs (*Zhou & Bao, 2020*). The expressions of 3,012 FerLncRNAs were found to be correlated (|R|>0.35 and *p* < 0.001) with the expression of FRGs. 77 FRGs were differentially expressed between benign and malignant tumor tissues.

## Enrichment analysis of differentially expressed FRGs in ccRCC

Based on the GO analysis at the biological process level, differentially expressed FRGs were mainly enriched in response to hypoxia, response to decreased oxygen levels, and response to oxygen levels; at the cellular component level, differentially expressed FRGs were mainly enriched in the apical part of the cell, apical plasma membrane, and basolateral plasma membrane; and at the molecular function level, differentially expressed FRGs were mainly enriched in iron ion binding, oxidoreductase activity, acting on single donors with the incorporation of molecular oxygen, and acting on NADPH (Fig. 1A). At these three levels, the top six enriched terms and the specific genes involved are presented in Fig. 1B.

Based on the KEGG pathway analysis, the differentially expressed FRGs were mainly enriched in the HIF-1 signaling pathway, ferroptosis, microRNA in cancer, the PPAR signaling pathway, bladder cancer, autophagy, programmed death ligand 1 (PD-L1) expression, and the PD-1 checkpoint pathway (Fig. 1C).

## Construction of the prognostic model in the training cohorts

The correlation between differentially expressed FerLncRNAs and patient survival information was evaluated using univariate Cox analysis in training cohorts, which identified 30 prognostic lncRNAs in the training cohort. Because of the large number, LASSO analysis was conducted to avoid overfitting of the model (Fig. 2A). Eventually, the following 10 FerLncRNAs were selected: LINC00894, DUXAP8, LINC01426, PVT1, MIR155HG , LINC01355, PELATON, LINC02609, MYG1-AS1, and PRKAR1B-AS1.

To further evaluate the significance of these differentially expressed FerLncRNAs in normal and malignant tissues and their prognostic significance in the model construction, multivariate Cox regression analysis was performed in the training group, which revealed that seven FerLncRNAs were independent prognostic factors for patients with ccRCC. Therefore, a seven-FerLncRNA signature was constructed to predict the OS of each patient with ccRCC. The risk score was calculated using the following equation:

$$Risk\ score = 0.967 \times Expression\ of\ LINC00894 + 0.786 \times Expression\ of\ DUXAP8 + 0.069$$
$$\times Expression\ of\ LINC01426 + 0.103 \times Expression\ of\ PVT1 + 0.132$$
$$\times Expression\ of\ PELATON + 0.245 \times Expression\ of\ LINC02609 + 0.291$$
$$\times Expression\ of\ MYG1 - AS1.$$

Based on the median risk score, patients were divided into a high- and low-risk group. Kaplan–Meier curves demonstrated that the high-risk group had a worse prognosis in the training cohort (Fig. 2B). The accuracy of the prognostic signature was evaluated by the ROC curve, the AUC values of 1, 3, 5, 7, and 10 years of overall survival (OS) were 0.896, 0.793, 0.801, 0.823, and 0.967, respectively (Fig. 2C). As to the 5-year AUC values of prognostic signature compared with other clinicopathological factors (age, gender, grade, stage, T, M, and N), the prognostic signature has the highest values (Fig. 2D). The heatmap showed remarkable differences in the expression of seven FerLncRNAs between the high-risk group and the low-risk group (Fig. 2E), the scatter plot indicated that ccRCC patients with a high-risk score had a lower survival rate than those with a low-risk score (Fig. 2F). Moreover, the distribution map of the risk score was consistent with the categorization of patient groups (Fig. 2G).

## Validation of the prognostic model

To validate the predictive capacity of the FRlncRNA signature, risk scores of patients were calculated in the testing cohorts and overall cohorts, and patients were classified into the low- risk group and the high-risk group based on the median risk scores in the training cohorts. The Kaplan–Meier survival analysis of OS in the testing and overall cohorts demonstrated that high-risk group had a worse prognosis (all $P < 0.001$, Figs. 3A and 3G). The 5-year ROC curves of testing cohorts (AUC = 0.739) and overall cohorts (AUC = 0.772) demonstrated that the FRlncRNA signature has a better predictive capability compared with other clinicopathological factors (Figs. 3B and 3H, respectively). The AUC values of 1-, 3-, 5-, 7-, and 10-years of OS were 0.599, 0.634, 0.739, and 0.838 in testing cohorts (Fig. 3C), and the the AUC values of 1-, 3-, 5-, 7-, and 10-years of OS were

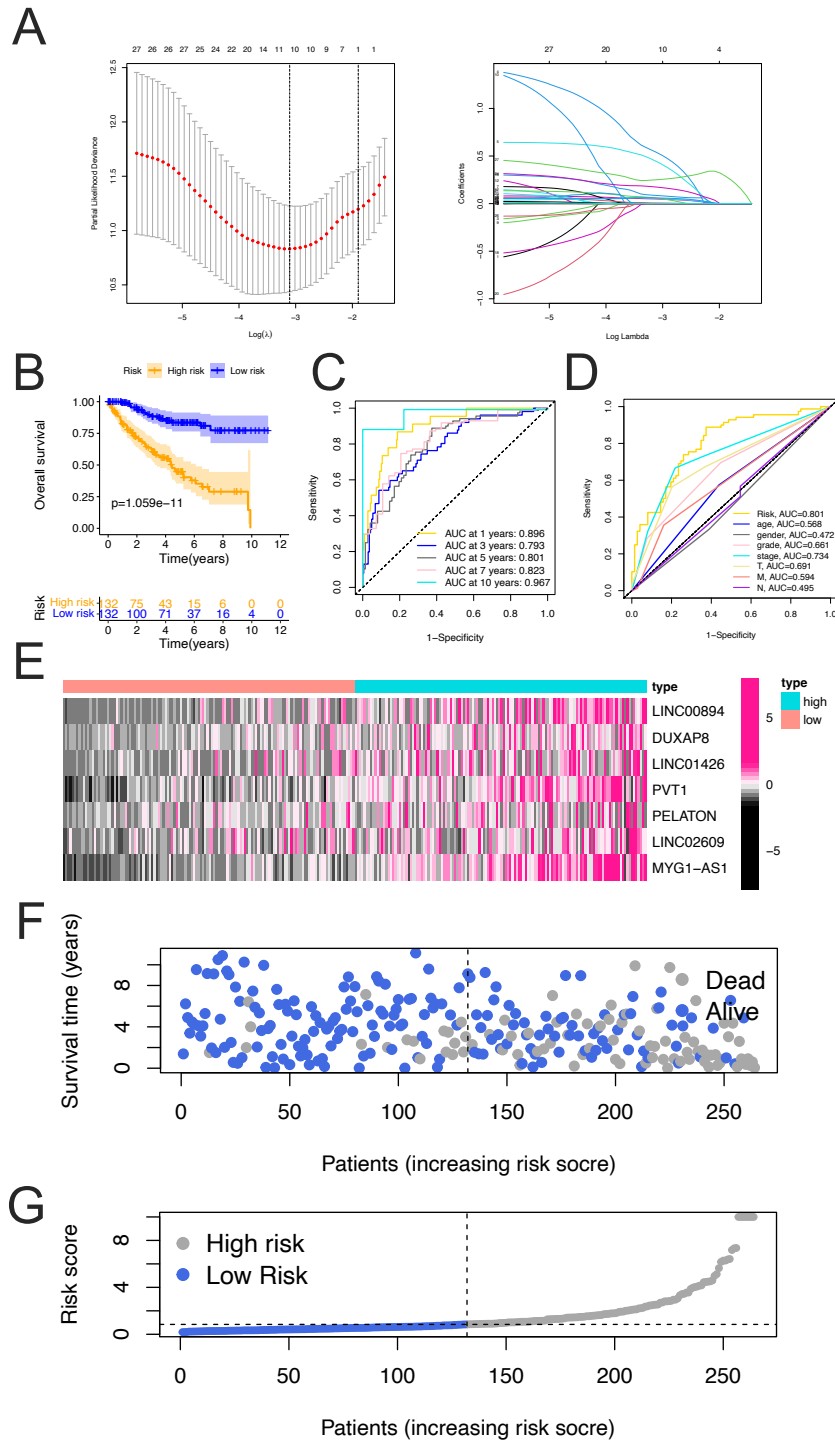

**Figure 2 Construction and evaluation of FerLncRNAs prognostic signature in training cohorts.** (A) Least absolute shrinkage and selection operator (LASSO) regression was performed with the minimum criteria. (B) Kaplan–Meier curves for overall survival (OS) of the patients from the high- and low-risk groups. (C) ROC curves and their AUC values represented (continued on next page...)

**Figure 2 (…continued)**
1-, 3-, 5-, 7, and 10-year predictions. (D) ROC curves of prognostic signature and clinicopathological factors (age, gender, grade, stage, T, M, and N) for 5-year AUC. (E) Heatmap showing expression of the seven FerLncRNAs between the high- and low-risk groups. (F) Scatter plot showing the correlation between the survival status and risk score of ccRCC patients. (G) Risk score distribution plot showing the distribution of high-risk and low-risk ccRCC patients.

0.739, 0.712, 0.772, and 0.909 in overall cohorts (Fig. 3I), these results further validate the prognostic signature. The consistent expression profiles of seven FRlncRNAs in the training cohorts are shown in the heatmaps (testing cohorts, Fig. 3D; overall cohorts, Fig. 3J). The survival rate of high-risk group was lower than that of low-risk group, and the risk score distribution map confirmed that the risk score of high-risk group was higher (testing cohorts, Figs. 3E and 3F; overall cohorts, Figs. 3K and 3L).

In addition, ICGC cohorts were used to evaluate the constructed model, which showed consistent expression profiles of the risk FRlncRNAs, and a good predictive capability of OS for patients with ccRCC (Fig. S2).

These results show that, compared with other prediction models reported in the recent studies with ferroptosis in ccRCC, our FRlncRNAs prediction model has great advantages and clinical operability with less LncRNA number and the highest 5-year AUC value (*Xing et al., 2021*; *Zhou et al., 2022*; *Chen et al., 2022d*; *Bai et al., 2022*) (Table S1). Therefore, it can be used as a good index to predict the prognosis of ccRCC patients.

Taken together, our data suggested that the FRlncRNA signature showed a stable prognostic-predictive power.

## PCA and stratified survival analysis of clinicopathological characteristics

The PCA schematic diagram shows two different risk levels of ccRCC patients in entire gene expression, ferroptosis genes expression, ferroptosis-related differentially expressed lncRNAs expression, and seven lncRNAs risk models (Figs. 4A–4D, respectively).

To assess the predictive ability of FRlncRNA signature and its stability in predicting OS in high-risk and low-risk groups, we performed stratified survival analysis of clinicopathological factors including age (<=60 years *vs.* >60 years), grade (Grade 1-2 *vs.* Grade 3-4), gender (Male *vs.* Female), stage (Stage I-II *vs.* Stage III-IV), T (T1-2 *vs.* T3-4), M (M1 *vs.* M0). The results of Kaplan–Meier survival analysis including different clinical factors further showed that OS in high-risk group was worse than that in low-risk group (all $P < 0.01$) (Fig. 4E).

## Correlation between the FerLncRNA prognostic signature and Clinicopathological features

Strong correlations were observed between the risk scores and clinicopathological characteristics (stage, grade, T, M, and survival status) with ccRCC (Fig. 5A), that is, as the stage, grade, metastasis and mortality increased, the risk score also gradually increased (Fig. 5B, all $P < 0.001$).

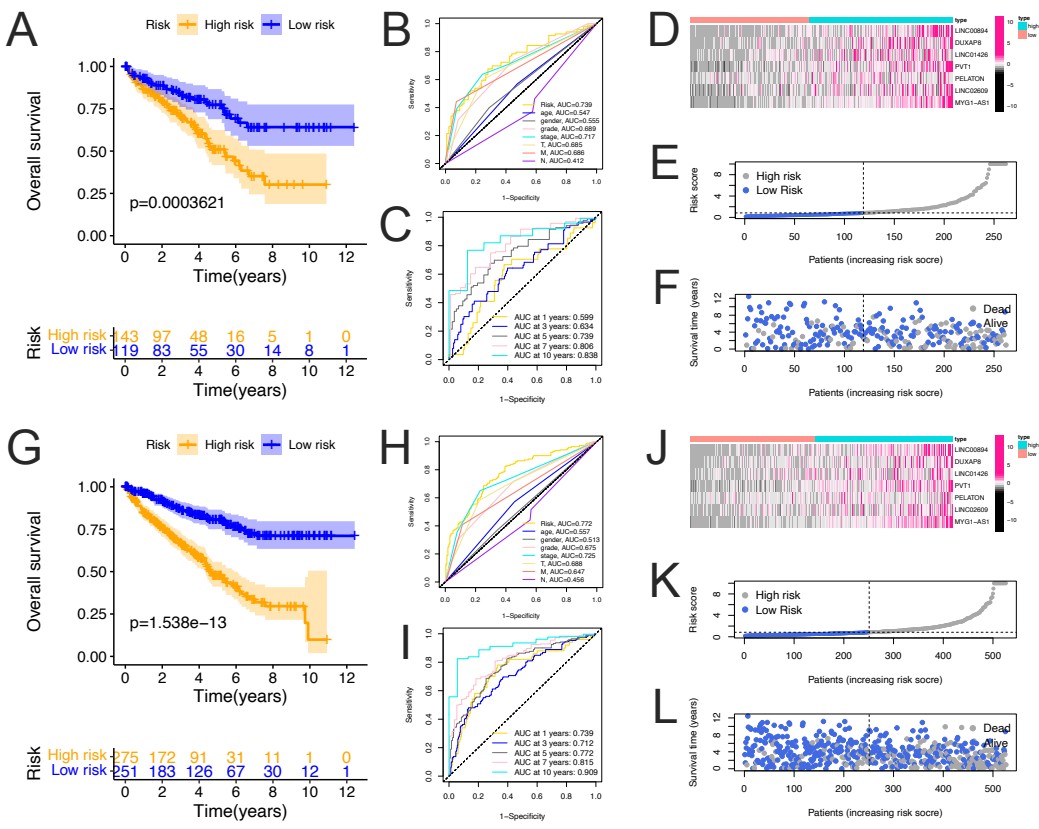

**Figure 3** **Validation of the FerLncRNAs prognostic signature for ccRCC patients in testing cohorts and overall cohorts.** Kaplan–Meier curves for overall survival (OS) of the patients from the high- and low-risk groups in the testing cohorts (A) and overall cohorts (G); ROC curves of prognostic signature and clinicopathological factors (age, gender, grade, stage, T, M, and N) for 5-year AUC in the testing cohorts (B) and overall cohorts (H); ROC curves and their AUC values showed 1-, 3-, 5-, 7-, and 10-year predictions in the testing cohorts (C) and overall cohorts (I); Heatmap of seven FRlncRNA expression profiles showed the expression of FRlncRNAs in high-risk and low-risk groups in the testing cohorts (D) and overall cohorts (J); Risk score distribution plot showed the distribution of high-risk and low-risk in the testing cohorts (E) and overall cohorts (K); Scatter plot showed the correlation between the survival status and risk score in the testing cohorts (F) and overall cohorts (L).

## Construction and evaluation of the prognostic nomogram

To determine whether the risk score was an independent prognostic factor in patients with ccRCC, univariate and multivariate Cox regression analyses were performed using clinical characteristics of the patients and their risk scores. The results demonstrated that the risk score was an independent prognostic factor ($P < 0.05$) (Figs. 6A and 6B). Then, using the clinicopathological characteristics, including age, grade, stage and risk score, the nomogram was constructed using the rms package in R to predict the 1-, 3-, 5-, and 10-year OS of patients with ccRCC (Fig. 6C). The results of the multivariate ROC curve showed that the 5-year AUC value of nomogram was 0.788, which was higher than that of the age (0.557), grade (0.675), and stage (0.725), indicating that the nomogram had the ability of accurate prediction for survival outcomes of ccRCC (Fig. 6D). The time-dependent AUC

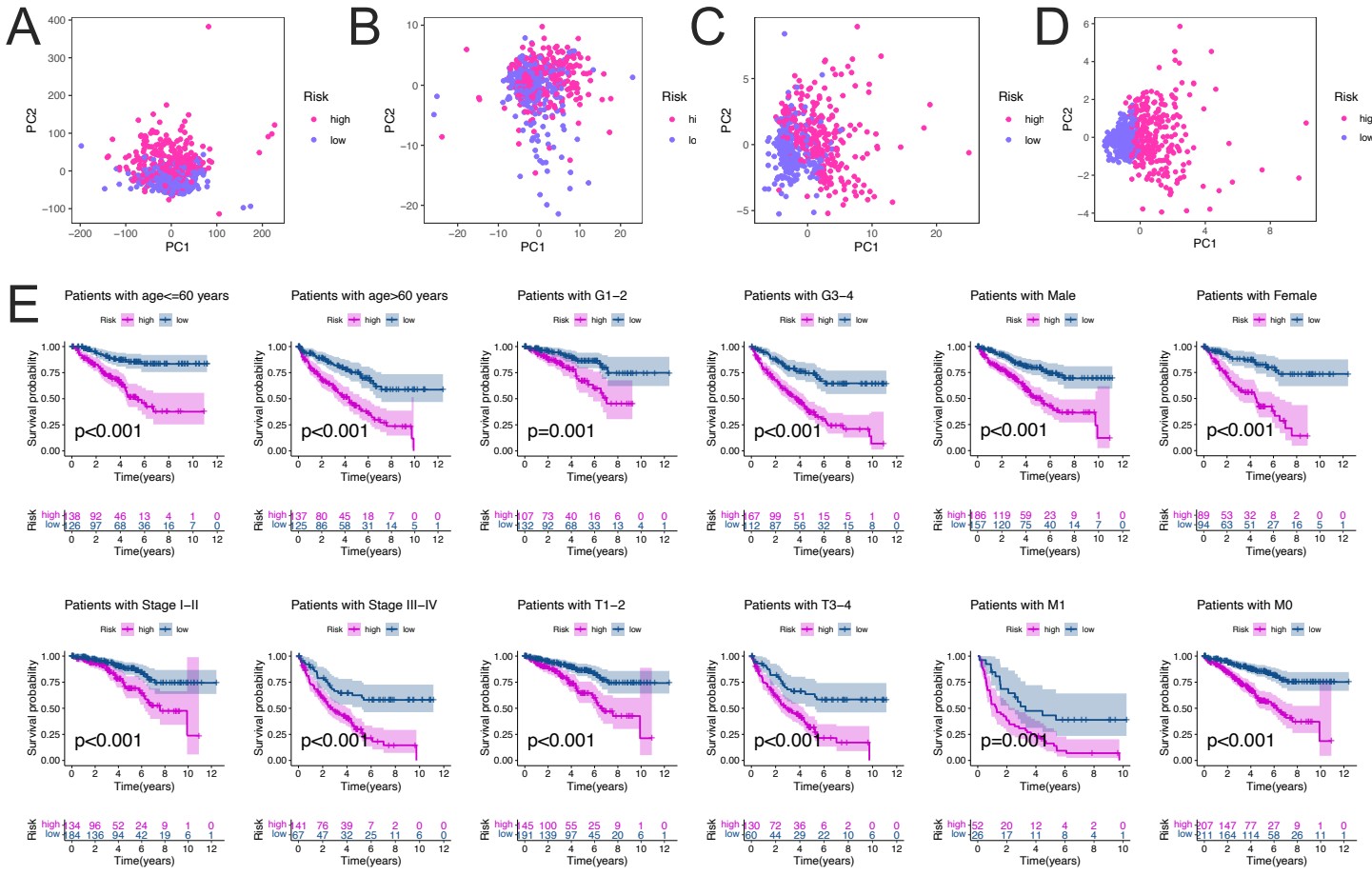

**Figure 4** **PCA analysis and survival analysis of ccRCC patients stratified by different clinicopathological features.** PCA analysis for the entire gene expression (A), Ferroptosis genes (B), differentially expressed ferroptosis-related lncRNAs (C), and seven FRlncRNAs risk models (D) in high and low risk groups in ccRCC patients. Kaplan–Meier curves (D) indicated the survival outcomes of high- and low-risk ccRCC patients stratified according to the age ($<=$60 years *vs.* >60 years), grade (Grade 1–2 *vs.* Grade 3-4), gender (Male *vs.* Female), stage (Stage I–II *vs.* Stage III–IV), T (T1-2 *vs.* T3-4), M (M1 *vs.* M0), respectively (all $p < 0.01$).

analysis showed that the prognostic value of the nomogram was significantly higher than that of age, stage, and grade over a time span of 1 to 10 years (Fig. 6E).

We used the calibration curve to observe whether the actual prognostic value was consistent with the predicted value of the nomogram and found that the calibration curves of 1-, 3-, 5-, and 10-year survival rates were consistent with the nomogram (Fig. 6F). The DCA curves also showed that the nomogram had a favorable prognostic effect and a better clinical value than stage (Fig. 6G). The clinical influences of the risk score for ccRCC patients in the training, and testing cohorts are showed in Fig. S3.

## Functional enrichment analysis of the risk signature

To explore the biological functions associated with the risk signature, the differentially expressed genes between the high- and low-risk groups were used to perform GO and KEGG analysis. GO analysis consisted of molecular function (MF) analysis mainly

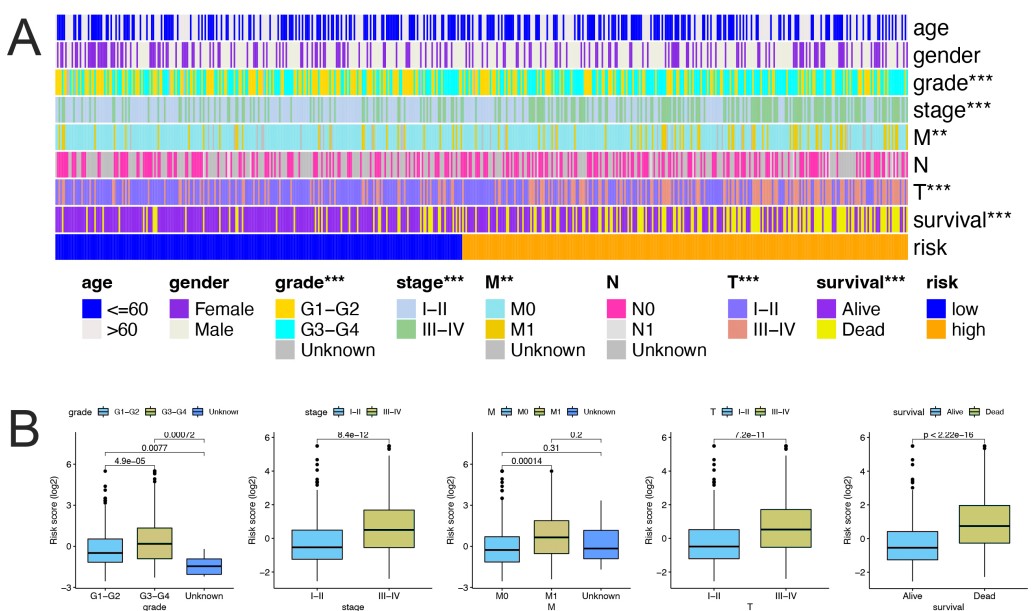

**Figure 5** **Relationships between the risk scores and clinicopathological features.** (A) Heatmap showing the FerLncRNA prognostic signature and clinicopathological features of the low- and high-risk patients with ccRCC. Color codes are used to indicate different clinicopathological parameters and risk levels. (B) Correlation analyses of the FerLncRNA prognostic signature with the clinicopathological characteristics of the patients with ccRCC according to the grade, stage, M, T, and survival status, respectively (all $P <$ 0.001).

including antigen biding, immunoglobulin receptor binding; cellular component (CC) analysis mainly containing immunoglobulin complex, external side of plasma membrane; biological process (BP) analysis mainly including B cell receptor signaling pathway, complement activation, and phagocytosis (Fig. 7A). The KEGG pathway enrichment analysis displayed that cytokine receptor interaction, IL-17 signaling pathway, and NF-kappa B signaling pathway were enriched (Fig. 7B). The "pathway-gene clustering" for GO (Fig. 7C) and KEGG enrichment analysis (Fig. 7D) were plotted.

## Construction of a lncRNA–mRNA coexpression network

We first explored the correlation between the seven FerLncRNAs, Fig. 8A shows that most of our FerLncRNAs are positively correlated with each other. There are 28 ferroptosis-related genes associated with seven FerLncRNAs. Figure 8B Sanky diagram indicates that there is a wide and complex correlation between them. To explore the potential roles of the seven FerLncRNAs in ccRCC, a lncRNA–mRNA coexpression network that contained 35 lncRNA–mRNA pairs was constructed using Cytoscape (Fig. 8C). The correlation of seven FerLncRNAs and 28 FRGs were plotted (Fig. 8D).

## External verification of the major genes

By inquiring the relevant literatures and online database (*Meng, Shao & Feng, 2021*; *Jiang et al., 2021b*; *Hu et al., 2020*; *Zhou et al., 2020*), we decided to select four FRlncRNAs (LINC00894, LINC01426, PVT1, and DUXAP8) for further investigation. In ICGC

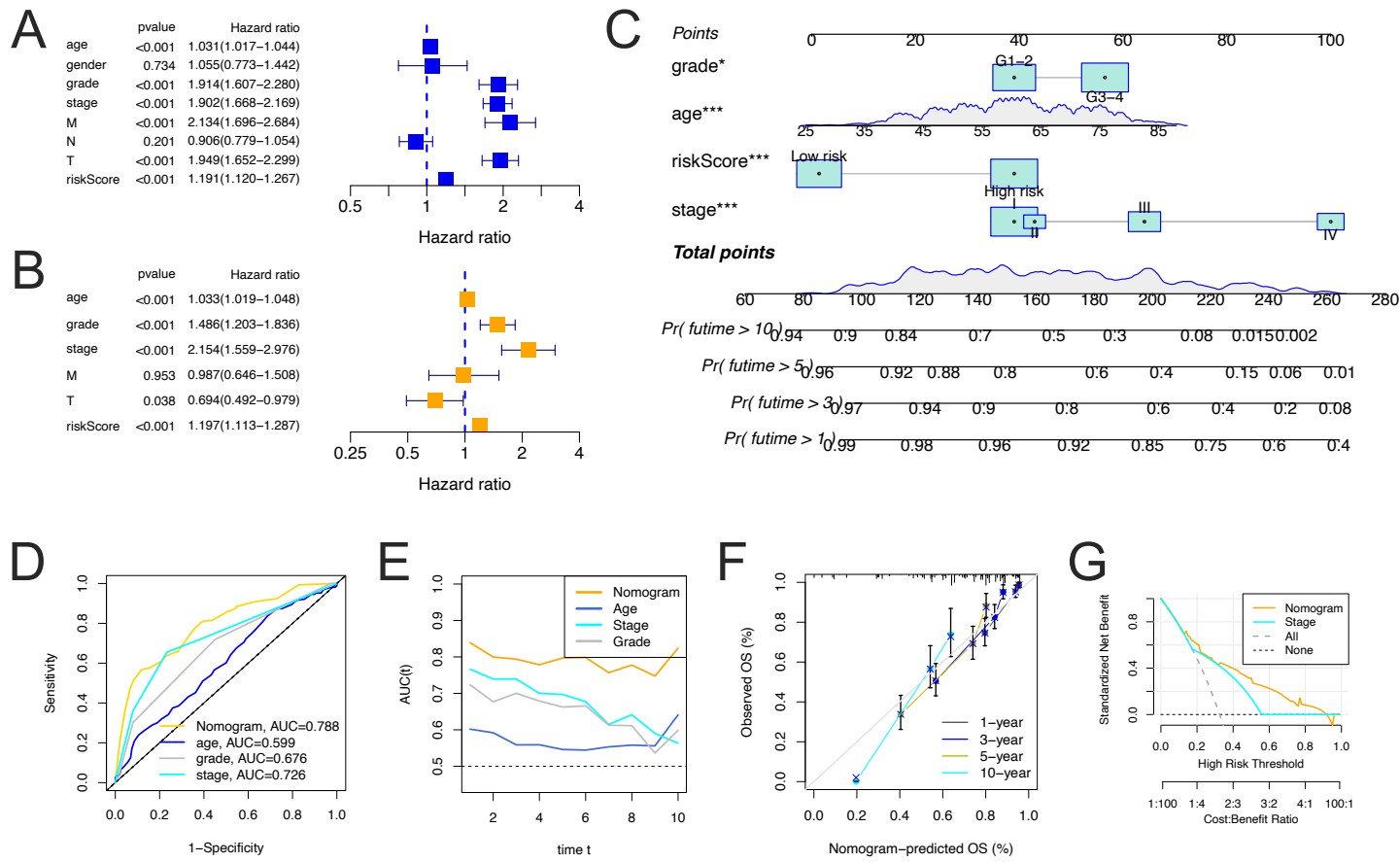

**Figure 6 Identification of independent prognostic variables and nomogram construction.** (A) Univariate Cox survival analysis showed that FerLncRNA signature and other clinical factors (age, gender, grade, stage, T, M, and N) were prognosis-related variables in overall cohorts. (B) Multivariate Cox survival analysis showed the FerLncRNA signature was an independent prognostic factor, as well as age, grade, and stage, T in overall cohorts. (C) Prognostic nomogram for predicting 1-, 3-, 5-, and 10-year OS of patients with ccRCC. (D) Multivariate 5-year ROC curve showed predictive accuracy of the nomogram was better to other clinicopathological variates. (E) Time-dependent ROC curves to compare AUC values of the nomograms and other clinical factors within a time range from 1 to 10 years. (F) Calibration curves of nomogram displayed the concordance between predicted and observed 1-, 3-, 5-, and 10-year OS. (G) DCA curves for the nomogram and stage.

database, the expression of LINC00894, LINC01426, PVT1, and DUXAP8 were obviously elevated in ccRCC compared with normal kidney tissues (Figs. 9A–9D, respectively, all $P < 0.05$). The expression trends were also observed as to LINC00894, LINC01426, and PVT1 which were further validated with ccRCC cohorts from GEO database (GSE15641, GSE46699, GSE40435, Figs. 9E–9G, respectively, all $P < 0.01$).

Further, we used two databases (GEPIA and K-M plotter) to explore four FRlncRNAs including expression levels, correlation with stage, and survival results. The expression levels of four FRlncRNAs were similar as validated by ICGC and GEO databases (Figs. 9H–9K). With the exception of LINC00894 (Fig. 9L), the expression levels of LINC01426, PVT1, DUXAP8 increased gradually with the increase of stages (Figs. 9M–9O, respectively, all $P < 0.05$), suggesting these FRlncRNAs were correlated with ccRCC progression. High expression levels of four FRlncRNAs were related to worse OS (Figs. 9P–9S, all $P < 0.05$).
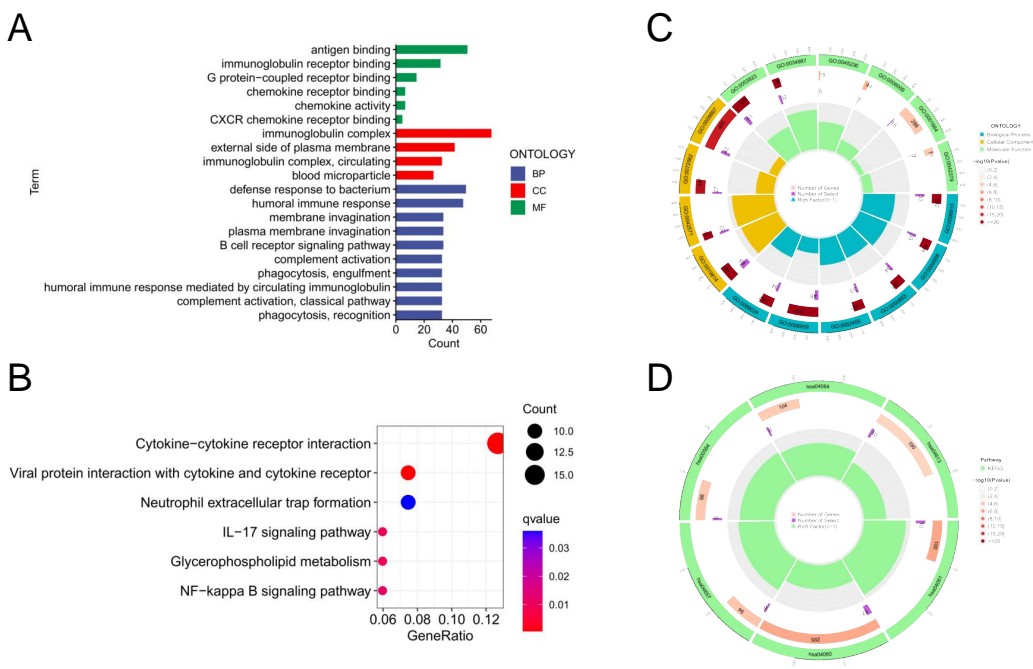

**Figure 7** **Functional enrichment analysis of seven risk lncRNAs.** (A) GO analysis including biological process (BP), cellular component (CC), and molecular function (MF); (B) KEGG pathway enrichment analysis. (C) Circos plot of the GO enrichment results. (D) Circos plot of the KEGG pathway enrichment results.

Similar results were also obtained in 530 ccRCC patients from the K-M plotter database (Figs. 9T–9W, all *P* < 0.01).

### *In vitro* experimental verification of the major genes

The expression levels of four FRlncRNAs were verified by qRT- PCR in the normal and tumor cells (Fig. 10, Table S2). The results showed that the overall trend in the expression levels of all four FRlncRNAs increased obviously in ccRCC cell lines (Caki-1, and 786-O) compared with normal renal proximal tubule epithelial cells (HK-2), which are consistent with our previous bioinformatics analysis based on public database.

### Immune landscape of the ccRCC microenvironment

Functional enrichment analysis suggested that a number of biological functions associated with the FerLncRNAs were involved in immune responses. Further, based on the results of immunotyping of pancancer in the literature (*Thorsson et al., 2018*), we compared the relationship between risk score and immunotyping of ccRCC, and found that there were significant differences between the existing immune subtypes C1, C2, C3, and C6 and risk score (Fig. 11A). Therefore, we consider that there is a potential correlation between our risk score and the immune infiltration response of ccRCC.

We further investigated the correlation of the risk score with the immune landscape of the ccRCC microenvironment. The high-risk group showed significantly higher immune, and ESTIMATE scores than those in the low-risk group (Fig. 11B). The heatmap of

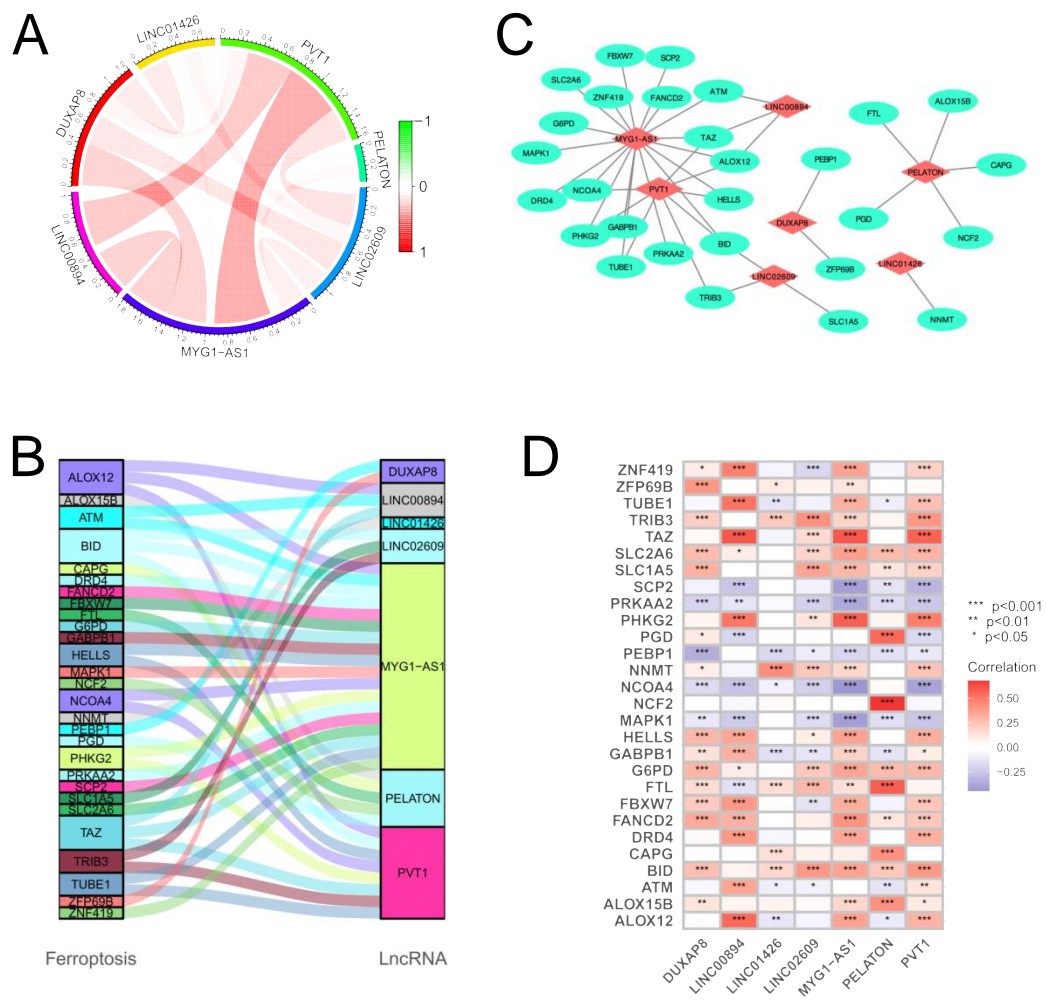

**Figure 8** **Co-expression network of the FRlncRNAs and FRGs.** (A) Circos plot showed the interaction relationship between seven FRlncRNAs; (B) Sankey diagram showing the degree of connection between the FerLncRNAs and ferroptosis-related genes. (C) Diagram of the FerLncRNA–mRNA interaction network. (D) Boxplot showed the relationship between seven FRlncRNAs and 28 FRGs.

immune responses based on the TIMER, CIBERSORT, CIBERSORT-ABS, QUANTISEQ, MCPCOUNTER, XCELL, and EPIC algorithms is shown in Fig. 11C; Based on this analysis, different algorithms indicated considerable differences between the two groups in terms of their immune infiltration functions. These findings fully confirmed that our FerLncRNA signature was strongly related to immune cell infiltration in ccRCC.

By ssGSEA, significant differences were observed between the high- and low-risk groups in terms of immune cells, including eosinophil, immature dendritic cell, mast cell, etc (Fig. 11D, all $P < 0.05$). Given the importance of checkpoint inhibitor-based immunotherapies, we further explored the differences in the expression of immune checkpoints between the two groups. Substantial differences were found in the expression of CD80, CD28, CTLA4, IDO2, PDCD1, and many other important indicators between the two groups of patients,

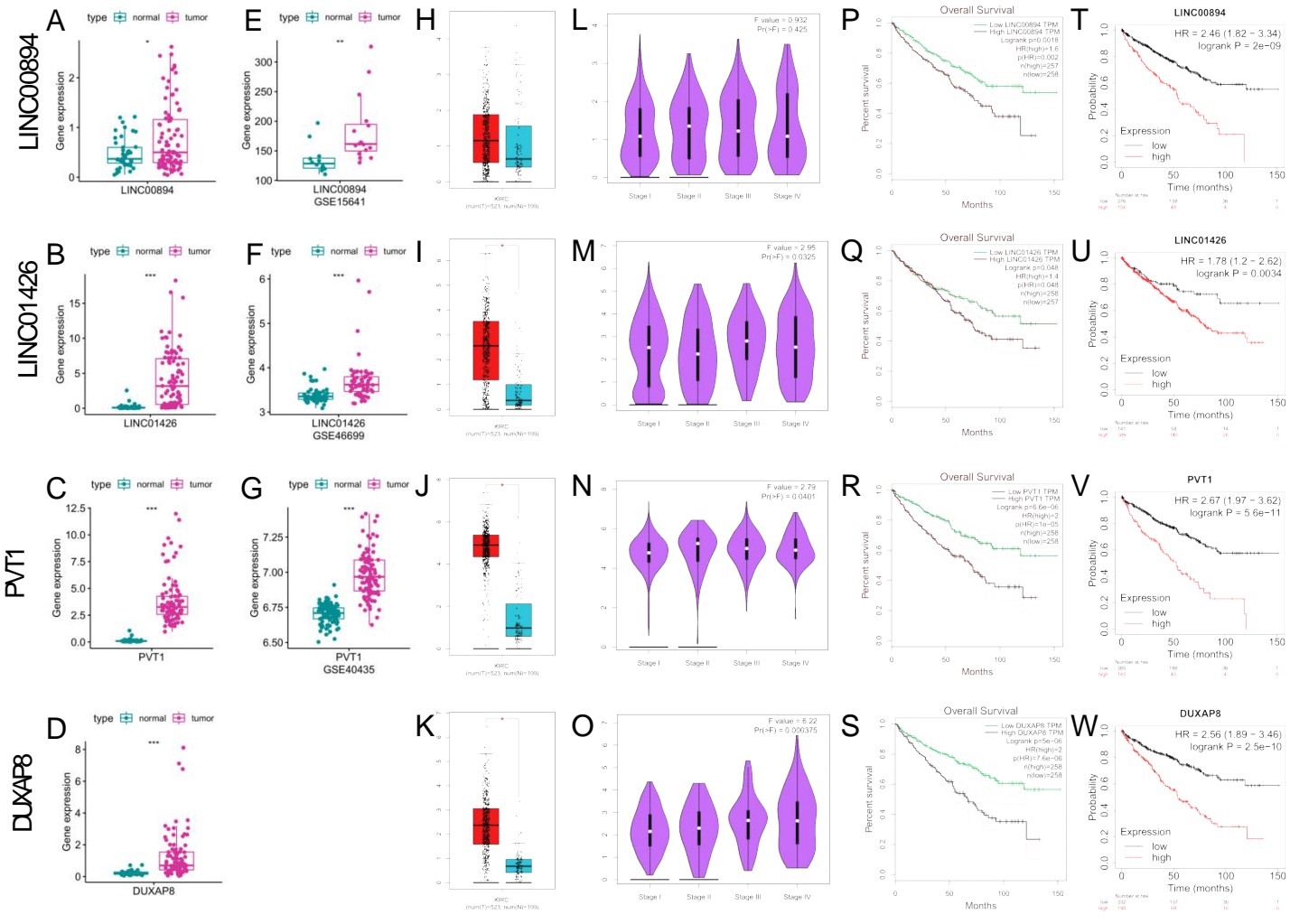

**Figure 9** **Verification of expression and prognosis of four FRlncRNAs from the ICGC, GEO, GEPIA and K-M Plotter databases.** The expression levels of four FRlncRNAs in ICGC database (A–D); The expression levels of LINC00894, LINC01426, and PVT1 in GSE15641, GSE46699, and GSE40435, respectively (E–G); The expression levels of four FRlncRNAs in GEPIA database (H–K); The correlation of expression levels of four FRlncRNAs with stage in GEPIA database (L–O); The Kaplan–Meier survival analysis of four FRlncRNAs of OS in GEPIA database (P–S); The Kaplan–Meier survival analysis of four FRlncRNAs of OS in K–M Plotter database (T–W).

nearly all these checkpoints were elevated in high-risk group (Fig. 11E, all $P < 0.05$). These results suggest that our risk score may have a potential correlation with the patient's response to immunotherapy.

## Predicting sensitivity of chemotherapy and response to immunotherapy in patients with ccRCC

We evaluated the relationship between the risk signature and the sensitivity to chemotherapy and targeted therapy drugs for ccRCC patients by the pRRophetic R package. Our results showed that, a significant difference was found between the two risk subgroups in the estimated IC50 values of 76 types of chemotherapy agents (including

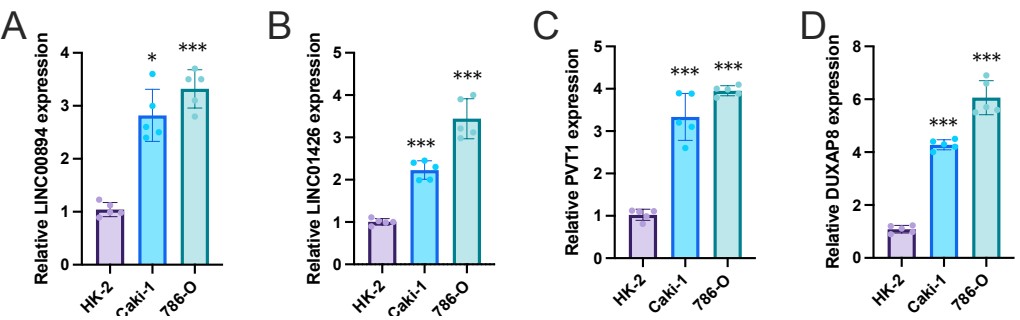

**Figure 10** **The expression levels of four FRlncRNAs *in vitro* by qRT-PCR.** qRT-PCR was used to measure the mRNA expression levels of LINC00894 (A), LINC01426 (B), PVT1 (C), and DUXAP8 (D) in human renal proximal tubule epithelial cells HK-2 and renal clear cell carcinoma cell lines (Caki-1, and 786-O). * $P < 0.05$, ** $P < 0.01$, *** $P < 0.001$.

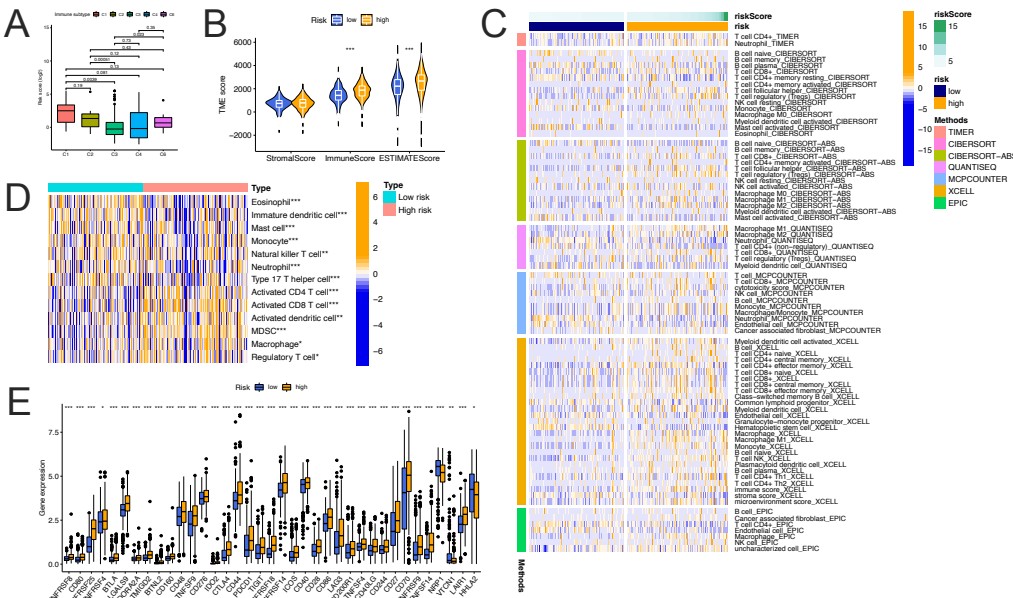

**Figure 11** **Correlation of the prognostic FerLncRNAs with the immune landscape of the ccRCC microenvironment.** (A) Difference in risk scores among the immune subgroups. (B) Comparison of immune, stromal, and ESTIMATE scores between the high- and low-risk groups. (C) Heatmap of immune responses based on the TIMER, CIBERSORT, CIBERSORT-ABS, QUANTISEQ, MCPCOUNTER, XCELL, and EPIC algorithms between the high- and low-risk groups. (D) Comparison of the ssGSEA scores between the high- and low-risk groups. (E) Expression of immune checkpoints between the high- and low-risk groups (* $p < 0.05$, ** $p < 0.01$, *** $p < 0.001$).

Etoposide, 5-Fluorouracil, Sorafenib, AKR inhibitor VIII, et al. all $p < 0.05$, Table 2). The IC50 values of DMOG, AKT inhibitor VIII, Ruxolitinib, Lapatinib, and Rapamycin were obviously lower in samples of the low-risk group than in those of the high-risk group (Table 2). However, interestingly, there are still 43 drugs with low expression of IC50 in the high-risk group, the high-risk group demonstrated much higher sensitivity

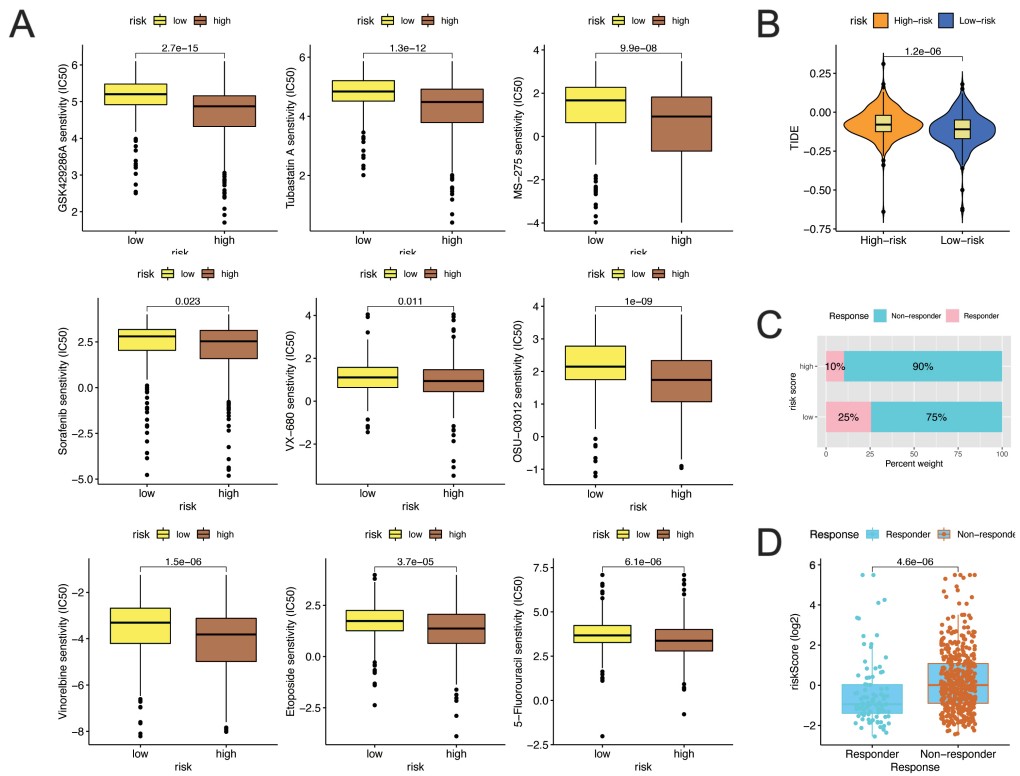

**Figure 12** **Predicting response to immunotherapy and sensitivity of chemotherapy in patients with ccRCC.** (A) Differences in drug sensitivity between the high-risk and low-risk groups based on IC50 values of GSK429286A, Tubastatin A, MS-275, Sorafenib, VX-680, OSU-03012, Vinorelbine, Etoposide, and 5-Fluorouracil between risk score and IC50 values. (B) TIDE score difference in high and low risk groups. (C) The differences in response results to immunotherapy between low-risk and high-risk groups by ImmuCellAI algorithm; (D) The scatter plot shows the correlation between immunotherapy responsiveness and risk score in ccRCC patients.

to the ROCK inhibitor (GSK429286A), HDAC inhibitor (Tubastatin A, MS-275), Raf inhibitor (Sorafenib), Aurora kinase inhibitor (VX-680), PDK-1 inhibitor (OSU-03012), Vinorelbine, Etoposide, 5-Fluorouracil than those of the low risk group (Fig. 12A). These results indicated that the risk score had potential predictive significance for chemotherapy and targeted therapy.

Finally, we evaluated the potential immunotherapy response in each patient by TIDE and ImmuCellAI algorithms. The results demonstrated that, patients in low-risk group were has better immunotherapy response (Fig. 12B), low-risk group patients were more likely to respond to immune checkpoint blockade (25%) than were patients in the high-risk group (10%) (Fig. 12C). In addition, the risk score was lower in the responders than in the non-responders (Fig. 12D). Taken together, these results indicated that the prognostic risk signature could predict the potential response to immunotherapy in ccRCC patients.

**Table 2   Risk score and chemotherapy drugs sensitivity in ccRCC patients.**

| Drug | Correlation Coefficient | P-value for correlation | IC50 value (median ± SD) | | P-value for sensitivity |
| | | | Low risk group | High risk group | |
|---|---|---|---|---|---|
| GSK429286A | −0.4116847 | 9.03E−23 | 5.202 ± 0.593 | 4.873 ± 0.81 | 2.75E−15 |
| Pyrimethamine | −0.4008079 | 1.22E−21 | 4.282 ± 0.925 | 3.537 ± 1.098 | 5.02E−15 |
| Tubastatin A | −0.3752999 | 7.17E−19 | 4.841 ± 0.655 | 4.487 ± 0.983 | 1.27E−12 |
| GNF-2 | −0.3697642 | 1.74E−18 | 2.509 ± 0.212 | 2.381 ± 0.277 | 2.36E−11 |
| OSU-03012 | −0.3339629 | 3.60E−15 | 2.148 ± 0.87 | 1.739 ± 0.957 | 1.02E−09 |
| AUY922 | −0.3162495 | 1.11E−13 | −3.092 ± 0.575 | −3.337 ± 0.566 | 3.26E−10 |
| BI-2536 | −0.3140472 | 1.67E−13 | −2.082 ± 0.534 | −2.324 ± 0.618 | 6.02E−08 |
| YM155 | −0.299257 | 2.41E−12 | −1.343 ± 6.914 | −3.863 ± 6.499 | 4.06E−07 |
| MS-275 | −0.2928415 | 8.04E−12 | 1.669 ± 1.596 | 0.922 ± 1.778 | 9.92E−08 |
| FTI-277 | −0.2830392 | 3.80E−11 | 2.161 ± 0.229 | 2.048 ± 0.286 | 2.43E−09 |
| CGP-60474 | −0.2801646 | 6.08E−11 | −2.236 ± 0.752 | −2.437 ± 0.688 | 1.05E−07 |
| WZ3105 | −0.2796805 | 9.98E−11 | 1.393 ± 1.891 | 0.666 ± 2.211 | 2.43E−06 |
| Vinorelbine | −0.2763082 | 1.13E−10 | −3.305 ± 1.291 | −3.819 ± 1.425 | 1.45E−06 |
| LAQ824 | −0.272809 | 1.98E−10 | −2.507 ± 0.854 | −2.854 ± 0.942 | 3.77E−07 |
| BMS-754807 | −0.2660241 | 5.69E−10 | 0.838 ± 0.636 | 0.546 ± 0.762 | 1.88E−07 |
| Phenformin | −0.2656272 | 6.05E−10 | 7.963 ± 0.91 | 7.547 ± 1.068 | 2.32E−06 |
| GW-2580 | −0.2635171 | 1.25E−09 | 5.759 ± 0.924 | 5.502 ± 1.334 | 2.09E−06 |
| NSC-207895 | −0.2633884 | 8.52E−10 | 4.27 ± 0.915 | 3.947 ± 0.989 | 1.34E−05 |
| Zibotentan | −0.2611228 | 1.20E−09 | 5.567 ± 0.209 | 5.478 ± 0.219 | 5.62E−07 |
| Salubrinal | −0.2593582 | 2.01E−09 | 4.363 ± 1.25 | 3.719 ± 1.492 | 7.17E−06 |
| CP466722 | −0.2555132 | 2.76E−09 | 3.095 ± 1.137 | 2.625 ± 1.333 | 3.18E−06 |
| NG-25 | −0.2477181 | 8.52E−09 | 3.007 ± 0.845 | 2.73 ± 1.088 | 1.07E−05 |
| JNK-9L | −0.2440906 | 1.42E−08 | −0.144 ± 0.531 | −0.34 ± 0.6 | 1.10E−05 |
| JW-7-24-1 | −0.2338737 | 5.75E−08 | 1.775 ± 0.72 | 1.479 ± 0.903 | 6.94E−06 |
| PHA-665752 | −0.2262784 | 1.56E−07 | 2.864 ± 0.212 | 2.798 ± 0.271 | 6.67E−06 |
| JQ12 | −0.2214814 | 2.88E−07 | 1.688 ± 1.216 | 1.353 ± 1.331 | 2.88E−05 |
| A-443654 | −0.2167294 | 5.21E−07 | −0.803 ± 0.312 | −0.871 ± 0.366 | 0.00069563 |
| Etoposide | −0.2154125 | 6.12E−07 | 1.737 ± 0.967 | 1.368 ± 1.227 | 3.73E−05 |
| Lisitinib | −0.2140062 | 7.45E−07 | 2.292 ± 0.59 | 2.001 ± 0.836 | 1.15E−06 |
| Mitomycin C | −0.198886 | 4.30E−06 | −0.598 ± 0.849 | −0.928 ± 0.934 | 9.40E−06 |
| 5-Fluorouracil | −0.1917897 | 9.45E−06 | 3.674 ± 1.028 | 3.368 ± 1.114 | 6.08E−06 |
| KIN001-135 | −0.1808101 | 3.03E−05 | 3.889 ± 0.253 | 3.793 ± 0.325 | 0.00061634 |
| Doxorubicin | −0.1787203 | 3.75E−05 | −1.719 ± 0.627 | −1.916 ± 0.81 | 0.00113553 |
| AZ628 | −0.1781504 | 0.00029364 | 5.71 ± 2.579 | 4.896 ± 2.837 | 0.00297656 |
| TAK-715 | −0.1702494 | 0.00010176 | 4.657 ± 1.055 | 4.349 ± 1.277 | 0.00130195 |
| Crizotinib | −0.1637402 | 0.00030632 | 3.094 ± 0.88 | 2.851 ± 1.197 | 0.01034253 |
| LY317615 | −0.160018 | 0.0002287 | 3.346 ± 0.612 | 3.082 ± 0.679 | 1.57E−05 |
| AS605240 | −0.1598432 | 0.00028584 | 3.831 ± 1.434 | 3.449 ± 1.679 | 0.00634062 |
| Sorafenib | −0.1433182 | 0.00118607 | 2.799 ± 1.447 | 2.534 ± 1.68 | 0.02333714 |

**Table 2** (*continued*)

| Drug | Correlation Coefficient | *P*-value for correlation | IC50 value (median ± SD) | | *P*-value for sensitivity |
| --- | --- | --- | --- | --- | --- |
| | | | Low risk group | High risk group | |
| CP724714 | −0.1393836 | 0.00150345 | 4.617 ± 0.785 | 4.516 ± 0.91 | 0.01613482 |
| TL-1-85 | −0.1369686 | 0.00170886 | 3.66 ± 1.051 | 3.47 ± 1.234 | 0.00884412 |
| VX-680 | −0.1223163 | 0.00496696 | 1.107 ± 0.813 | 0.936 ± 0.998 | 0.01145153 |
| GW843682X | −0.1101968 | 0.01143801 | −2.886 ± 1.025 | −2.965 ± 1.119 | 0.02462889 |
| GSK1904529A | 0.10348973 | 0.01758529 | 2.261 ± 0.283 | 2.287 ± 0.289 | 0.02385087 |
| CGP-082996 | 0.12168176 | 0.00532862 | 1.451 ± 3.49 | 1.893 ± 3.837 | 0.04197339 |
| Dasatinib | 0.136895 | 0.0018856 | −0.708 ± 3.034 | 0.201 ± 3.198 | 0.01358105 |
| Z-LLNle-CHO | 0.13805049 | 0.00150446 | 0.263 ± 1.627 | 0.403 ± 1.507 | 0.01753678 |
| Rapamycin | 0.15665587 | 0.00031024 | −2.749 ± 1.604 | −2.523 ± 1.713 | 0.00827571 |
| Lapatinib | 0.15866254 | 0.0002588 | 2.268 ± 0.832 | 2.431 ± 0.835 | 0.00580935 |
| Shikonin | 0.16179519 | 0.00019418 | −0.997 ± 3.32 | −0.591 ± 3.634 | 0.00529048 |
| XMD8-85 | 0.16653263 | 0.00014891 | 0.466 ± 8.166 | 1.222 ± 8.87 | 0.00678214 |
| Ruxolitinib | 0.17062802 | 8.53E−05 | 3.906 ± 0.472 | 4.039 ± 0.556 | 0.0114337 |
| Midostaurin | 0.17622863 | 4.82E−05 | −0.858 ± 2.128 | −0.446 ± 2.277 | 0.00351609 |
| BX-912 | 0.177726 | 4.15E−05 | 2.573 ± 1.062 | 2.858 ± 1.306 | 0.00592782 |
| Obatoclax Mesylate | 0.18180522 | 2.73E−05 | −1.836 ± 2.237 | −1.316 ± 2.363 | 0.00139064 |
| GSK-650394 | 0.18184426 | 2.72E−05 | 3.063 ± 0.854 | 3.357 ± 1.044 | 0.00263797 |
| KIN001-102 | 0.18465183 | 2.03E−05 | 2.466 ± 0.682 | 2.636 ± 0.839 | 0.00197243 |
| Bryostatin 1 | 0.18612326 | 2.38E−05 | −3.542 ± 1.523 | −3.184 ± 1.709 | 0.00529447 |
| XL-184 | 0.18723462 | 1.54E−05 | 2.219 ± 1.298 | 2.493 ± 1.425 | 0.00124207 |
| WZ-1-84 | 0.1894432 | 1.27E−05 | 3.504 ± 1.102 | 3.763 ± 1.103 | 0.00109396 |
| QS11 | 0.19118071 | 1.01E−05 | 2.886 ± 0.968 | 3.157 ± 1.181 | 0.00151151 |
| Bleomycin | 0.19127995 | 1.02E−05 | 0.006 ± 3.981 | 0.776 ± 4.472 | 0.00150329 |
| AS601245 | 0.1931847 | 8.11E−06 | 1.962 ± 0.409 | 2.066 ± 0.438 | 0.00075864 |
| CMK | 0.19339315 | 7.93E−06 | 1.154 ± 2.384 | 1.664 ± 2.546 | 0.00048235 |
| Parthenolide | 0.19787401 | 5.13E−06 | 2.076 ± 1.924 | 2.608 ± 2.121 | 0.00108445 |
| Saracatinib | 0.20334878 | 9.68E−06 | 1.193 ± 2.263 | 2.109 ± 2.329 | 0.00163244 |
| NSC-87877 | 0.2054239 | 3.32E−06 | 4.11 ± 1.689 | 4.579 ± 1.875 | 0.00287139 |
| FR-180204 | 0.2073988 | 1.61E−06 | 4.913 ± 0.32 | 4.971 ± 0.349 | 0.00070296 |
| Pazopanib | 0.20879199 | 1.36E−06 | 2.636 ± 1.269 | 2.982 ± 1.427 | 0.00019755 |
| FMK | 0.21931092 | 3.78E−07 | 4.629 ± 0.954 | 4.927 ± 1.068 | 0.00014473 |
| XMD14-99 | 0.22055458 | 7.39E−07 | 3.562 ± 1.773 | 4.14 ± 1.951 | 0.00092496 |
| CAL-101 | 0.2394262 | 3.25E−08 | 3.432 ± 1.979 | 4.083 ± 2.134 | 5.39E−05 |
| Thapsigargin | 0.24828123 | 7.87E−09 | −5.281 ± 3.865 | −4.504 ± 4.747 | 4.48E−06 |
| ZSTK474 | 0.25063554 | 5.61E−09 | 0.436 ± 1.358 | 0.964 ± 1.538 | 1.49E−05 |
| AKT inhibitor VIII | 0.2598575 | 1.45E−09 | 2.277 ± 0.286 | 2.367 ± 0.286 | 1.41E−07 |
| DMOG | 0.27301586 | 1.91E−10 | 5.339 ± 1.995 | 6.165 ± 2.518 | 5.66E−07 |
## DISCUSSION

Ferroptosis is a nonapoptotic form of programmed cell death. The main causes of ferroptosis are iron-dependent accumulation of reactive oxygen species (ROS) and the consumption of plasma membrane polyunsaturated fatty acids. A normal nucleus, increased membrane density, a ruptured outer membrane, and atrophic or deficient mitochondria are morphological features of cellular ferroptosis (*Jiang et al., 2021a*). When the concentration of ROS exceeds the elimination capacity of the antioxidant system, ROS can oxidize unsaturated fatty acids in the cell membrane to form lipid peroxides, which can damage the structure and function of cells directly or indirectly (*Xie & Guo, 2021*). In recent years, the deregulation of ferroptosis was associated with numerous human pathologies, including cancer (*Ganini et al., 2022*). Ferroptosis can promote tumorigenesis and cancer progression by inducing gene mutations and epithelial–mesenchymal transition and implicating other mechanisms  (*Tang et al., 2021*). It is reported that kidney cancer shows high susceptibility to ferroptosis (*Ganini et al., 2022*; *Chen et al., 2022c*). As a result, ferroptosis-related biomarkers may be useful as both diagnostic and therapeutic targets for ccRCC. An in-depth understanding of these biomarkers is expected to provide a breakthrough in the molecular mechanism of ferroptosis-related ccRCC initiation and development.

In recent years, it has become clear that lncRNAs play a role in cell differentiation, cell cycle regulation, stem cell pluripotency, and the maintenance of various biological processes, such as nerve growth differentiation and tumorigenesis (*Fang & Fullwood, 2016*). lncRNAs play complex roles in oncogenesis as oncogenes and tumor repressors (*Goodall & Wickramasinghe, 2021*). lncRNAs localize to chromatin, interact with proteins and target RNAs, and promote cancer phenotypes by forming proliferation circuits, tumor suppressor circuits, viability circuits, motility circuits, and cross-talks between different mechanisms (*Fang & Fullwood, 2016*; *Schmitt & Chang, 2016*). Many lncRNAs have been implicated in the occurrence and development of urological tumors (*Zuo et al., 2022*), and some of them are involved in the regulation of ferroptosis, such as in bladder cancer, lncRNA RP11-89 could induce tumorigenesis and reduce the accumulation of cellular iron by sponging the miR-129-5p/PROM2 pathway, thus leading to ferroptosis inhibition (*Luo et al., 2021*); in prostate cancer, lncRNA OIP5-AS1 could act as a ceRNA that sponges miR-128-3p to increase the level of SLC7A11 (*Zhang et al., 2021*), lncRNA PCAT1 could be activated by TFAP2C to suppress ferroptosis by interacting with c-Myc (*Jiang et al., 2022*); in ccRCC, some ferroptosis-associated lncRNAs (such as LUCAT1, LINC02027, LINC00460, etc.) could become prognostic signatures (*Han et al., 2022*; *Xing et al., 2021*; *Zhou et al., 2022*).

Our research was based on the screening of TCGA data for FRG-related lncRNAs to find differentially expressed lncRNAs that could be used for the prognosis of patients with ccRCC. Survival analysis allowed the construction of a seven-lncRNA-based prediction model that could strongly predict the prognosis of patients with ccRCC. We selected four lncRNAs (LINC00894, LINC01426, PVT1, and DUXAP8) for further investigation by ICGC, GEO, GEPIA, K-M plotter databases, and qRT-PCR. Currently, LINC00894,

LINC01426, PVT1, and DUXAP8 have been reported to be related to oncogenesis and development. LINC00894 enhances cell proliferation and invasion by binding with miR-429 to mediate ZEB1 expression in breast cancer (*Meng, Shao & Feng, 2021*); what's more, in thyroid cancer, its increased expression reduces the oncogenic properties by sponging let-7e−5p to promote TIA-1 expression (*Chen et al., 2022a*). LINC01426 modulates CTBP1/miR-423-5p/FOXM1 axis via interacting with IGF2BP1 to aggravate ccRCC progression (*Jiang et al., 2021b*), it also aggravates the malignant progression through miR-661/Mdm2 axis in glioma (*Shu et al., 2022*), it serves as a prognostic indicator in lung adenocarcinoma by triggering growth and metastasis (*Deng et al., 2022*), what's more, in a recently reported study on ferroptosis-related lncRNAs of glioma, LINC01426 was identified to be related to ferroptosis, its knockdown could significantly increase in the $Fe^{2+}$ levels and the erastin-induced ROS levels in glioma cells (*Huang et al., 2022*). LncRNA DUXAP8 acts as an oncogene in most tumors, its abnormal overexpression is associated with the proliferation, invasion, migration, anti-autophagy, and poor prognosis of tumors (*Wang et al., 2022a*; *Wang et al., 2022b*), it may act as a potential therapeutic target for cancer (*Wang et al., 2022b*); in a research on identification of necroptosis-related lncRNAs in hepatocellular carcinoma, DUXAP8 was identified as a prognostic lncRNA and related to patients' prognosis (*Chen et al., 2022b*); In addition to our research, other studies have also found that it is involved in the process of ferroptosis in tumor, such as: it can act with ferroptosis-associated gene FANCD2 to form DUXAP8-miR-29c-FANCD2 axe in hepatocellular carcinoma (*Yang et al., 2022*), it was also identified as one of ferroptosis-related lncRNAs in kidney carcinoma (*Xing et al., 2021*). Similar to DUXAP8, lncRNA PVT1 was considered as involved in the ferroptosis progress, it may regulate ferroptosis through miR-214-mediated TFR1 and TP53 expression (*Lu, Xu & Lu, 2020*), it was identified as one of the ferroptosis-related lncRNAs to construct a panel for predicting tumor progression, microenvironment (*He et al., 2021*), and radiotherapy response (*Zheng et al., 2021*) in glioma; in liver cancer, PVT1 is involved in the regulatory mechanism of lncPVT1/miR-214-3p/GPX4 axis and plays a role in ketamine suppressing the viability of liver cancer cells and inducing ferroptosis.

Studies on the other lncRNAs also proved their functions in oncogenesis, especially in the ferroptosis process, such as knockdown of lncRNA PELATON enhanced sensitivity to ferroptosis inducers to inhibit cell proliferation and invasion in glioblastoma cells (*Fu et al., 2022*); LINC02609 was positively correlated with late stage, grade, and distant metastasis in ccRCC (*Su et al., 2021*), and associated with OS in soft tissue sarcoma (*He et al., 2017*), what's more, other similar research also identified LINC02609 as ferroptosis-related lncRNA in kidney carcinoma (*Han et al., 2022*; *Xing et al., 2021*; *Bai et al., 2022*; *Shu et al., 2021*). Further research on the these lncRNAs, especially how these lncRNAs participate in the process of ferroptosis, can provide new insights for revealing the mechanism of renal cell carcinoma at the molecular level.With the in-depth study of tumor immunology and molecular biology, immunotherapy provides a new perspective for tumor treatment. At present,the immune checkpoint inhibitors used to treat advanced renal cell carcinoma, such as Keytruda and Opdivo, can enhance the immune response against renal cell carcinoma by blocking PD-1. However, in contrast to most other types of anti-PD-1 responsive

solid tumors, a high infiltration by CD8+ T cells in ccRCC patients has been previously associated with a worse prognosis (*Fridman et al., 2017*; *Braun et al., 2020*). This is different from the common paradigm that the tumor is infiltrated by a large number of immune cells, forming an "infiltrated" or "hot" environment in the tumor, which will better respond to PD-1 blockade (*Braun et al., 2020*; *Chen & Mellman, 2017*). This result was also observed in our study, that is, the prognosis of patients in the high-risk group was poor, but their immune function and immune checkpoint gene expression were increased, the predictive immunotherapy response by TIDE and ImmuCellAI showed that the low-risk groups patients could be benefited. Furthermore, by with the pRRophetic algorithm, we identified 76 types of chemotherapy agents which could be useful in treatment of ccRCC patients.

There are some limitations in this study. First, our research was based on retrospective data available in the TCGA public database, this FerLncRNA prediction model and its clinical utility need to be further verified using multicenter, prospective, real-world data. Second, our research only revealed the relationship between FerLncRNAs and the TIME, while potential mechanisms and specific clinical applications need to be further explored experimentally.

### Funding
The work was supported by the Guide project for the Natural Science Foundation of Liaoning Province (No. 2019-ZD-0796), the 345 Talent Project of Shengjing Hospital of China Medical University. The funders had no role in study design, data collection and analysis, decision to publish, or preparation of the manuscript.

### Grant Disclosures
The following grant information was disclosed by the authors:
Guide project for the Natural Science Foundation of Liaoning Province: 2019-ZD-0796.
345 Talent Project of Shengjing Hospital of China Medical University.

### Competing Interests
The authors declare there are no competing interests.

### Author Contributions
- Lincheng Ju conceived and designed the experiments, performed the experiments, analyzed the data, authored or reviewed drafts of the article, and approved the final draft.
- Yaxing Shi performed the experiments, prepared figures and/or tables, and approved the final draft.
- Gang Liu conceived and designed the experiments, analyzed the data, authored or reviewed drafts of the article, and approved the final draft.

## Data Availability

The code is available in the Supplementary Files.

## Supplemental Information

Supplemental information for this article can be found online at http://dx.doi.org/10.7717/peerj.14506#supplemental-information.

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
