# Peer review of "Identification and validation of a ferroptosis-related lncRNA signature to robustly predict the prognosis, immune microenvironment, and immunotherapy efficiency in patients with clear cell renal cell carcinoma"

_PeerJ, doi:10.7717/peerj.14506_

## Round 0.1 · original submission · Major Revisions

Dear Dr. Liu,

Thank you for submitting your manuscript " Identification and validation of a ferroptosis-related lncRNA signature to robustly predict the prognosis, immune microenvironment, and immunotherapy efficiency in patients with clear cell renal cell carcinoma" to PeerJ. We have now sufficiently received reports from three reviewers. After careful consideration, we have decided to invite a major revision of the manuscript.

As you will see from the reports copied below, the reviewers raise important concerns regarding the lack of bioinformatic methodology, clinical validation in an independent cohort, and statistical evaluation of the study for eg. AUC analysis. We find that these concerns limit the strength of the study, and therefore we ask you to address all of the reviewers' comments with additional work. Without substantial revisions, we will be unlikely to send the paper back for review.

Important:
If you feel that you are able to comprehensively address the reviewers’ concerns, please provide a point-by-point response to these comments along with your revision. Please show all changes in the manuscript text file with track changes or color highlighting. If you are unable to address specific reviewer requests or find any points invalid, please explain why in the point-by-point response.

Best regards,

Abhishek Tyagi, PhD
Academic Editor,
PeerJ

Reviewer 1 ·

Basic reporting

1 The review of related work is not sufficiently thorough and not sufficiently specific. The authors should cite references if appropriate.
2 The authors should double-check their texts throughout the manuscript because there are some minor errors, such as the Figure 4E legend and ‘similar’ in L369.

Experimental design

1 The computational details are not mentioned, such as ssGSEA in L171, and the authors should provide enough information about which software, libraries, parameters, etc., were used to implement their analysis.
2 The authors should provide a sample size for their analysis. For example, the sample size for their qRT-PCR analysis.
3 The authors should provide enough information to replicate their results. For example, in L381-L382, how they did immune subtyping according to ref19 and what the immune subtypes and samples information they use.

Validity of the findings

No comment.

·

Basic reporting

In the submitted manuscript Ju et al. presented their results of primarily bioinformatic re-analyses of publicly available data on ferroptosis-related genes and lncRNAs in clear cell renal cell carcinoma (ccRCC) and assessment of their prognostic significance.
Unfortunately, the quality of English language is not very high, there are lots of wrong or incomprehensive sentences or phrases throughout the text. For example, majority of "Cell Culture and Treatment" and "RNA Extraction, Reverse Transcription, and Quantitative Real-time PCR (qRT-PCR)" subsections weren't written in standard, understandable English language. Therefore, this manuscript would need comprehensive professional English proofreading.
At several places in the manuscript authors have stated that "impact of ferroptosis on lncRNAs-dependent ccRCC progression, immune microenvironment and immunotherapy response has been scarcely studied" without citing any particular examples, i.e., studies.
Majority of figures are generic outputs of R packages and in smaller resolution most text would not be readable.
Although authors have backed most of their hypotheses at first sight, the stringency of statistical procedures with whom they reached their conclusions is not satisfactorily, i.e., it is very low.

Experimental design

Unfortunately, authors have wrongfully performed several very basic steps in designing their study, performing analyses, explaining obtained results, and what is most important, drawing conclusions:
1) Majority of their general findings were based on correlation analyses ("Screening for
FerLncRNAs was performed using Pearson's correlation analysis with the reported
ferroptosis-related genes."), because of which authors misplaced correlation for causation. And that was even performed very poorly, because setting corr. coef. to be meaningful if |R| > 0.35 is very poor judgement because, for instance, according to Mukkaka (PMID: 23638278) this means very low correlation!
2) Another big part of authors' findings (and conclusions) were based on meaningless AUC values. Authors should inspect, for instance, Mandraker's paper (PMID: 20736804) to se how to properly interpret AUC values, since majority of theirs actually suggest no or very low discrimination power.
3) In vitro experiments performed on two, almost randomly selected, ccRCC cell lines really do not bring any added value to their bioinformatic (theoretical) results. It is suggested that authors should perform some actual functional assays to find real molecular mechanisms behind the roles of those lncRNAs in the etiopathology of ccRCC.
4) Authors' "validation" results were obtained with GEPIA and K-M plotter software which are based on the same TCGA-KIRC dataset which was used as training and test datasets of their primary bioinformatic analyses, so thus obtained validation results actually do not mean real validation!
In addition, this study is to a great deal irreproducible since explained bioinformatic methods lack important details so anyone could repeat them.

Validity of the findings

As stated previously, findings presented in this manuscript were built on very loose base. They weren't actually validated because same set of samples (TCGA-KIRC dataset) were repeatedly used in multiple software. In addition, validation of lncRNAs expression in two ccRCC cell lines was actually meaningless and worthless.
Taken all together, presented conclusions weren't actually corroborated by obtained results.
In addition, first half of "Discussion" is just a summary of obtained results.

Additional comments

For all used datasets (TCGA-KIRC, GSE15641, GSE46699, and GSE40435) authors should provide original references.
For all used R packages authors should provide their version numbers and cite proper references if published in scientific journals.
For all web-based tools and databases authors should provide VALID URLs and cite their references.
All bioinformatic procedures should be explained in enough detail (which particular dataset and software were used) so anyone could repeat those analyses.
For every bioinformatic procedure used dataset should be precisely stated.

Reviewer 3 ·

Basic reporting

In this study, the authors established a ferroptosis-related lncRNA prognostic signature to predict the individual prognosis of ccRCC. The study investigated its efficiency and accuracy in the training, testing, total cohorts in TCGA database, as well as ICGC database. The results obtained with GEPIA, and K-M Plotter data supported the predictive ability of the major lncRNAs in risk signature. In addition, immune cell infiltration and check point expressions associated with this signature were explored. Althouge these studies are informative, a variety of issues need to be resolved.

1. The language needs to be improved, where some proper nouns were inaccurately expressed.
2. There was too little connection between the results and ferroptosis in the discussion, please increase the description between the them.
3. For row 121 and other parts, a total of 259 FRGs will be obtained from FerrDb Database, but how did you get 239 FRGs ?
4. For row 122, FerrDb Database was wrongly expressed.
5. For Fig.3A , it should be training cohort not overall training.

Experimental design

1. AUC <0.8 doesn’t indicate strong prognostic ability.
2. For Fig.12B, the TIDE value was higher in the high-risk group, which contradicted Fig.11D. Could you please explain it?

Validity of the findings

no comment

---

## Round 0.2 · accepted · Accept

Dear Dr. Chen,

We are delighted to accept your manuscript, entitled "Identification of prognostic factors and nomogram model for patients with advanced lung cancer receiving immune checkpoint inhibitors," for publication in PeerJ. Thank you for choosing to publish your interesting work with us.


With kind regards,
Abhishek Tyagi
Academic Editor, PeerJ

Reviewer 1 ·

Basic reporting

The authors have addressed all of my concerns and the manuscript has been improved a lot.

Experimental design

The authors have addressed all of my concerns and the manuscript has been improved a lot.

Validity of the findings

No comment.

Additional comments

No comment.